# Room-Temperature Synthesis of Highly-Efficient Eu^3+^-Activated KGd_2_F_7_ Red-Emitting Nanoparticles for White Light-Emitting Diode

**DOI:** 10.3390/nano12244397

**Published:** 2022-12-09

**Authors:** Yongqiang Zhong, Qian Wu, Jiujun Zhu, Peiqing Cai, Peng Du

**Affiliations:** 1Department of Microelectronic Science and Engineering, School of Physical Science and Technology, Ningbo University, Ningbo 315211, China; 2College of Optical and Electronic Technology, China Jiliang University, Hangzhou 310018, China

**Keywords:** white-LED, Eu^3+^ doping, KGd_2_F_7_, luminescence

## Abstract

Luminescent materials with high thermal stability and quantum efficiency are extensively desired for indoor illumination. In this research, a series of Eu^3+^-activated KGd_2_F_7_ red-emitting nanoparticles were prepared at room temperature and their phase structure, morphology, luminescence properties, as well as thermal stability, have been studied in detail. Excited by 393 nm, the resultant nanoparticles emitted bright red emissions and its optimal status was realized when the Eu^3+^ content was 30 mol%, in which the concentration quenching mechanism was triggered by electric dipole–dipole interaction. Through theoretical analysis via the Judd–Ofelt theory, one knows that Eu^3+^ situates at the high symmetry sites in as-prepared nanoparticles. Moreover, the internal and extra quantum efficiencies of designed nanoparticles were dependent on Eu^3+^ content. Furthermore, the studied nanoparticles also had splendid thermal stability and the corresponding activation energy was 0.18 eV. Additionally, via employing the designed nanoparticles as red-emitting constituents, a warm white light-emitting diode (white-LED), which exhibits low correlated color temperature (4456 K), proper luminous efficiency (17.2 lm/W) and high color rendering index (88.3), was developed. Our findings illustrate that Eu^3+^-activated KGd_2_F_7_ nanoparticles with bright red emissions are able to be used to promote the performance of white-LED.

## 1. Introduction

Luminescent compounds doped with rare-earth ions, which present the satisfactory capacity of converting incident light to various emissions, have been widely explored, because of their potential application to many diverse fields, such as pressure monitoring, optical thermometry, white light-emitting diode (white-LED), energy conversion, etc. [1,2,3,4,5]. Nowadays, there is considerable attention paid to phosphor-converted white-LED, which has been widely employed to replace conventional lamps, since it possesses plenty of advantages, including small size, high-energy conversion, high brightness, long-working lifetime, environmental friendliness, etc. [6,7]. To achieve the commercial phosphor-converted white-LED, a facile route, namely, the Y_3_Al_5_O_12_:Ce^3+^ yellow-emitting phosphors being excited by a blue chip, was adopted [8,9]. However, on account of the shortage of red constituents in the luminescence profile, the emitted white light showed a low color rendering index (CRI < 80) and high correlated color temperature (CCT > 7000 K), which was not beneficial for its vivid application [10,11]. Stimulated by these features, researchers tried to obtain warm white light by utilizing a commercial near-ultraviolet (NUV) chip to excite the mixed-monochrome (i.e., blue–green–red) phosphors [12,13]. Evidently, developing single-emitting phosphors, especially red-emitting compounds, is extremely important in order to modify the performance of phosphor-converted white-LED. 

Eu^3+^, as one of the rare-earth ions, is an outstanding red-emitting activator because of its unique emissions originating from ^5^D_0_ → ^7^F_J_ (J = 0,1,2,3,4) transitions [14,15]. As we know, the luminescence characteristics of rare-earth ions are chiefly decided by the intrinsic performance of the host materials, where materials with small phonon energies are most favorable [16,17]. Thus, finding a proper host material is the most facile strategy to realize highly efficient rare-earth ions activated luminescent materials. Compared with other inorganics, fluorides have been widely studied as luminescent hosts for rare-earth ions. Specifically, ALn_2_F_7_-type (A = Li, Na, K; Ln = Y, Gd, Lu, Sc, etc.) fluorides have been regarded as a new type of good fluoride hosts for rare-earth ions [18,19,20]. Guo and his co-workers reported that the Tm^3+^/Yb^3+^-co-doped NaY_2_F_7_ nanocrystals not only exhibited intense blue emissions, but also were promising candidates for optical thermometry [21]. Xia and his co-workers found that the color-controllable emissions (i.e., from yellow to white light) were able to be realized in Dy^3+^/Sm^3+^-co-doped KGd_2_F_7_ nanocrystals [22]. Although some good results were obtained in rare-earth ions activated ALn_2_F_7_-type fluorides, greater effort is still required to further explore their luminescence properties, as well as their potential applications. 

In this study, Eu^3+^ and KGd_2_F_7_ were selected as the dopant and luminescent hosts, respectively, to prepare Eu^3+^-activated KGd_2_F_7_ nanoparticles. The crystal structure, morphology, luminescence properties, luminescent dynamic process, thermal quenching behaviors and quantum efficiency of studied samples were inspected in detail. Moreover, via the utilization of the Judd–Ofelt theory, the environmental behaviors of Eu^3+^ in KGd_2_F_7_ host lattices have been theoretically discussed. In addition, a warm white-LED was fabricated by using the designed nanoparticles, so as to identify their promising application to solid-state lighting. 

## 2. Materials and Methods

### 2.1. Synthesis of Eu^3+^-Activated KGd_2_F_7_ Nanoparticles at Room Temperature

A convenient chemical precipitation reaction technology was employed to prepare the KGd_2-2*x*_F_7_:2*x*Eu^3+^ (KGd_2_F_7_:2*x*Eu^3+^; *x* = 0.05, 0.10, 0.15, 0.20, 0.30, 0.40, 0.50 and 0.60) nanoparticles at room temperature. The raw materials were KNO_3_, Gd(NO_3_)_3_·6H_2_O, Eu(NO_3_)_3_·6H_2_O and NH_4_F, which were all bought from the Aladdin Company (Shanghai City, China), and their purities were 99%, 99.9%, 99.99% and 98%, respectively. The proper amount of KNO_3_, Gd(NO_3_)_3_·6H_2_O and Eu(NO_3_)_3_·6H_2_O powders were weighed through employing an electronic balance, and then solution one was gained by transferring them into 10 mL of ethylene glycol (EG; Aladdin Company, Shanghai City, China). Meanwhile, solution two was achieved by putting NH_4_F (7 mmol) into 25 mL of EG. After that, these two solutions were mixed with each other by means of stirring. After stirring for 2 h, the designed products were obtained through the following processes of centrifugation: washing with water and ethanol, and heating at 80 °C for 2 h.

### 2.2. Characterization

The crystal structure and morphology of the final nanoparticles were examined via the use of an X-ray diffractometer (Bruker D8 Advance, Cu Kα radiation, wavelength is 1.5406 Å, Bremen, Germany) and a field-emission scanning electron microscope (FE-SEM; HITACHI SU3500, Tokyo, Japan). The excitation and emission spectra of the designed nanoparticles were recorded through a fluorescence spectrometer (FS5, Edinburgh, UK), in which the surrounding temperature of nanoparticles was adjusted by a heating platform (Linkam HFS600E-PB2, Salfords, UK). Next, to measure the excitation spectrum, an optical filter with the cutoff wavelength of 510 nm (i.e., λ ≤ 510 nm), was adopted, whereas a cutoff filter (λ ≥ 400 nm) was employed to record the emission spectrum. Via the utilization of a fluorescence spectrometer (FSL1000, Edinburgh, UK), the decay time and luminescence efficiency of the resultant nanoparticles were tested. A multichannel spectroradiometer (SPEC-3000A; Measurefine; Hangzhou City, China) was applied to characterize the electroluminescence (EL) features of the developed white-LED.

## 3. Results

### 3.1. Phase Structure and Morphology

Figure 1a shows the X-ray diffraction (XRD) patterns of KGd_2_F_7_:2*x*Eu^3+^ nanoparticles. Clearly, the diffraction profiles of resultant nanoparticles were similar, and they did not only match well with the previous reported result [18], but also perfectly coincided with those of the standard KGd_2_F_7_ (JCPDS#270387), manifesting that the final nanoparticles had a pure monoclinic phase and Eu^3+^ entered into the KGd_2_F_7_ host lattices. Moreover, the positions of diffraction peaks shifted slightly to smaller angles; this was caused by the different ionic sizes between doping ions (Eu^3+^) and replaced ions (Gd^3+^), as is illustrated in Figure 1b, further indicating that Eu^3+^ was incorporated into the KGd_2_F_7_ host lattices by occupying the Gd^3+^ site. 

In order to check the morphology evaluation of resultant products, the FE-SEM graphs of KGd_2_F_7_:2*x*Eu^3+^ (0.05 ≤ *x* ≤ 0.60) nanoparticles were measured and demonstrated in Figure 2a–h. Evidently, uniformly nano-sized particles were observed in all of the prepared samples in which their sizes were around 23 nm. Additionally, as Eu^3+^ content increased, the size and shape of the yield particles remained unchanged, as is displayed in Figure 2a–h, manifesting that the morphology of designed compounds were independent on Eu^3+^ doping. Thereby, the morphology of studied samples had little impact on their luminescence properties. Furthermore, these elements (i.e., K, Gd, F and Eu) presented in developed nanoparticles were uniformly distributed over the whole particles, as is demonstrated in Figure 2i–m. 

### 3.2. Luminescence Behaviors and Thermal Stability

For the purpose of inspecting the luminescence behaviors of designed nanoparticles, the luminescence spectra of typical KGd_2_F_7_:0.60Eu^3+^ nanoparticles were recorded and displayed in Figure 3a. The excitation spectrum, in which the monitoring wavelength was 593 nm, consisted of many narrow peaks, in which their central wavelengths were 362, 376, 382, 393, 414 and 463 nm, arising from the absorption of Eu^3+^ from ^7^F_0_ to ^5^D_4_, ^5^G_2_, ^5^G_3_, ^5^L_6_, ^5^D_3_ and ^5^D_2_ levels, respectively [23,24]. Among these excitation bands, the sharp peak located at 393 nm exhibited the strongest intensity, illustrating that the commercial NUV chip was able to be employed as the excitation lighting source for the developed nanoparticles. Thus, we selected the excitation wavelength to measure the emission spectrum. As shown in Figure 3a, when the excitation wavelength was 393 nm, there were five sharp peaks in the recorded luminescence spectrum, in which their central wavelengths were 554, 593, 613, 650 and 700 nm, corresponding to the intra-4f transitions of Eu^3+^, that is, ^5^D_1_ → ^7^F_0_, ^5^D_0_ → ^7^F_1_, ^5^D_0_ → ^7^F_2_, ^5^D_0_ → ^7^F_3_ and ^5^D_0_ → ^7^F_0_, respectively [23,24]. From the previous literature, one knows that these two emissions originating from ^5^D_0_ → ^7^F_1_ and ^5^D_0_ → ^7^F_2_ transitions, which belong to magnetic dipole (MD) and electric dipole (ED) transitions, respectively, are the featured emissions of Eu^3+^ [24,25]. Specifically, the MD transition is scarcely impacted by the surrounding environment and its intensity is stronger than other emissions when Eu^3+^ is located at a high symmetry position, whereas the ED transition is sensitive to the surrounding environment and it prevails in the luminescence profile when Eu^3+^ takes up the low symmetry site [24,25]. It is shown in Figure 3a that the emission intensity of the MD transition was stronger that of the ED transition, implying that Eu^3+^ takes the high symmetry site in designed nanoparticles, which will be further discussed in the following section. 

Generally, the content of rare-earth ions has a huge impact on the fluorescent intensities of phosphors. In order to seek out the best doping content of Eu^3+^ in KGd_2_F_7_ host lattices, the KGd_2_F_7_:2*x*Eu^3+^ nanoparticles were developed, and their luminescence properties were recorded, as shown in Figure 3b. Excited by 393 nm, all of the synthesized nanoparticles displayed the characteristic emission bands of Eu^3+^ and their positions were hardly changed by altering Eu^3+^ content. However, as Eu^3+^ content increased, the emission intensity was significantly changed, and its maximum value was obtained at *x* = 0.30. When Eu^3+^ content was over 30 mol%, concentration –quenching occurred and the emission intensity showed a downward tendency, as shown in Figure 3c. In the case of the concentration quenching of Eu^3+^, it was caused by two different mechanisms of electric multipole interaction and exchange interaction, and they were able to be identified by estimating its critical distance (*R_c_*). According to previous works, *R_c_* value between dopants can be achieved with the aid of the following formula, based on the critical doping content (*x_c_*) [26]:(1)$$R_{c}=2{\left(\frac{3V}{4\pi x_{c}Z}\right)}^{1/3}$$
where *Z* denotes the quantity of cation sites in the studied samples and *V* refers to the volume of unit cell. In this work, the values of *x_c_*, *Z* and *V* for synthesized nanoparticles were 0.30, 2 and 78.11 Å^3^, respectively [18]. As a consequence, it is known that the *R_c_* value of Eu^3+^ in synthesized nanoparticles was around 6.29 Å, and was significantly greater than 5 Å, manifesting that the electric multipole interaction prevailed over the concentration quenching mechanism in the final products. Furthermore, to fully comprehend the involved concentration quenching mechanism, the following function was adopted to study the relation between fluorescence intensity (*I*) and doping concentration (*x*) [27]:(2)log(I/x)=C−θ/3×log(x)
where *C* is coefficient and *θ* exhibits three diverse values of 6, 8 and 10 corresponding to dipole–dipole, dipole–quadrupole and quadrupole–quadrupole interactions, respectively. Based on the recorded emission spectra, plot of log(*I*/*x*) versus log(*x*) (presented in Figure 3d) was used to evaluate the *θ* value. These experimental data were linearly fitted and the slope (−*θ*/3) was −2.29 (see Figure 3d). Thereby, the *θ* value was estimated to be 6.87, which is closer to 6 than 8, stating that the involved concentration quenching mechanism in designed nanoparticles pertained to an electric dipole–dipole interaction. Note that the emission intensity of the ^5^D_0_ → ^7^F_1_ transition (593 nm) was always stronger than that of the ^5^D_0_ → ^7^F_2_ transition (613 nm) in the resultant nanoparticles, suggesting that the symmetry properties of the sites taken by Eu^3+^ in KGd_2_F_7_ host lattices were hardly changed by altering the doping content. Additionally, the International Commission on Illumination (CIE) coordinate of KGd_2_F_7_:0.60Eu^3+^ nanoparticles was determined to be (0.672,0.372) located at the edge of the red region. This revealed that the designed nanoparticles were able to emit visible red emissions with high color purity, which is further confirmed by the optical images excited by NUV (400 nm) light, as shown in Figure 3e. 

Room-temperature decay curves of KGd_2_F_7_:2*x*Eu^3+^ nanoparticles were measured to further investigate the luminescence kinetic, as shown in Figure 4a. When excited at 393 nm and monitored at 593 nm, these measured decay profiles were able to be fitted via the use of a single exponential decay model:(3)$$I\left(t\right)=I_{0}+A\exp\left(-t/\tau \right)$$
where, *I*(*t*) is the fluorescence intensity at time *t*, while *I*_0_ is the fluorescence intensity at times *t* much longer than the decay time *τ*, and *A* is ascribed to constant. Accordingly, the lifetimes of Eu^3+^ in studied samples were found to be around 8.46, 6.88, 5.95, 544, 4.45, 3.14, 2.09 and 1.42 ms, respectively, when the Eu^3+^ content was 5, 10, 15, 20, 30, 40, 50 and 60 mol% (see Figure 4b). It is clear that the lifetime exhibited a rapid downward tendency as Eu^3+^ content increased, which suggests that the concentration quenching existed in the developed nanoparticles. Furthermore, the relation between decay time and dopant content was able to be further analyzed by Aizel’s model [28]:(4)$$\tau \left(x\right)=\tau_{0}/\left[1+\frac{x}{C_{0}}*exp\left(-N/3\right)\right]$$
where *τ*(*x*) refers to the lifetime at dopant content of *x*; *τ*_0_ is assigned to the radiative transition lifetime; *C*_0_ is constant and *N* denotes the generated phonon number for quenching the studied level by means of the cascade multi-phonon process. Through fitting these experimental data by Equation (4), the value of *τ*_0_ was found to be about 11.55 ms. 

As stated above, Eu^3+^ takes up the high symmetry site in KGd_2_F_7_ host lattices. To theoretically confirm this conclusion, it was necessary to evaluate the optical transition arguments of Ω_λ_ (λ = 2, 4, 6) via the utilization of the Judd–Ofelt theory. Since all of the nanoparticles showed the same luminescence profile, the KGd_2_F_7_:2*x*Eu^3+^ (*x* = 0.30) nanoparticles were selected to calculate the Ω_λ_ value. On the basis of the previous literature [29,30], it is widely known that the integrated emission intensity and total radiative transition rate meet the following relation:(5)$$I=\underset{J=0,1,2,3,4,6}{\sum }I_{J}=\alpha \underset{J=0,1,2,3,4,6}{\sum }A_{J}$$
where *I_J_* and *A_J_* are assigned to the fluorescence intensities and radiative transition probability of ^5^D_0_ → ^7^F_J_ (J = 0, 1, 2, 3, 4, 6) transitions, respectively, and *α* refers to constant. Notably, with the help of following formula and radiative transition lifetime, the exact value of total radiative transition probability was able to be estimated [29,30]:(6)$$\underset{J=0,1,2,3,4,6}{\sum }A_{J}=\frac{1}{\tau_{0}}$$
From Figure 4b, it is evident that the *τ*_0_ value of ^5^D_0_ level of Eu^3+^ in KGd_2_F_7_:0.60Eu^3+^ nanoparticles was 11.55 ms. As a result, the value of *α* was revealed to be 18,503.8. On the other hand, the MD transition rate can be expressed through using the following formula, proposed by Judd and Ofelt [31,32]:(7)$$A_{J\to J^{\prime }}^{MD}=\frac{64\pi^{4}v^{3}}{3h\left(2J+1\right)}n^{3}S_{MD}$$
where *h* refers to the Planck constant with an invariable value of 6.626 × 10^−27^ erg s; *v* is the wavenumber of *J* → *J*′ transition; *n* represents the refractive index of prepared nanoparticles and *S_MD_* denotes the MD strength, where its value is 7.83 × 10^−42^ [29,30]. Herein, only the ^5^D_0_ → ^7^F_1_ transition belongs to the MD transition, and thus, *n* value of KGd_2_F_7_:0.60Eu^3+^ nanoparticles was demonstrated to be approximately 1.39. Furthermore, in accordance with the Judd–Ofelt theory, the ED transition rate could also be achieved [31,32]:(8)$$A_{J\to J^{\prime }}^{ED}=\frac{64\pi^{4}e^{2}v^{3}}{3h\left(2J+1\right)}\frac{n{\left(n^{2}+2\right)}^{2}}{9}\underset{\lambda =2,4,6}{\sum }\Omega_{\lambda }\langle \psi J\left|U^{\lambda }\right|\psi^{\prime }J^{\prime }\rangle$$
where *e* = 4.8 × 10^−10^ esu is ascribed to elementary charge; $\langle \psi J\left|U^{\lambda }\right|\psi^{\prime }J^{\prime }\rangle$ is the reduced matrix element for the *J* → *J*′ ED transition, in which its value is 0.0032 and 0.0023, respectively, when the ED transition is ^5^D_0_ → ^7^F_2_ and ^5^D_0_ → ^7^F_4_ [30,33]. Since the emission originating from the ^5^D_0_ → ^7^F_6_ ED transition was not probed in the designed nanoparticles (see Figure 3a), the Ω_6_ value could not be calculated. Consequently, the values of Ω_2_ and Ω_4_ of Eu^3+^ in the resultant nanoparticles were determined to be 1.06 × 10^−20^ and 2.18 × 10^−20^ cm^−2^, respectively. Universally, Ω_2_ is highly dependent on the surrounding environment and its value will be large when the dopant occupies the low symmetry site, whereas Ω_4_ is only determined by the bulk properties and the rigidity of host materials [29,30]. Obviously, the Ω_4_ value was much larger than Ω_2_ value, indicating that Eu^3+^ occupied the high symmetry site in KGd_2_F_7_ host lattices.

In order to further characterize the luminescence properties of developed nanoparticles, the quantum efficiencies of KGd_2_F_7_:2*x*Eu^3+^ nanoparticles were measured under 393 nm excitation. The measured spectra of KGd_2_F_7_:0.60Eu^3+^ nanoparticles as a representative sample are shown in Figure 4c. In this work, the following equations were adopted to investigate the internal and quantum efficiencies [15,34]:(9)$$\mathrm{IQE}=\int L_{S}/\left(\int E_{R}-\int E_{S}\right)$$
(10)$$\mathrm{AE}=\left(\int E_{R}-\int E_{S}\right)/\int E_{R}$$
(11)EQE=AE×IQE
where IQE and EQE are the internal and extra quantum efficiencies, respectively; AE is the absorption of the excitation light; *L_S_* stands for the emission spectrum of designed compounds; *E_S_* and *E_R_* are assigned to the integrated intensities of excitation light with and without developed nanoparticles in integrating sphere, respectively. Excited by 393 nm, the IQE values of KGd_2_F_7_:2*x*Eu^3+^ nanoparticles were determined to be 42.6%, 58.0%, 73.6%, 78.1%, 78.4%, 59.5%, 43.6% and 20.2%, respectively, when *x* = 0.05, 0.10, 0.15, 0.20, 0.30, 0.40, 0.50 and 0.60, whereas their EQE values were 6.0%, 8.3%, 9.4%, 13.4%, 16.1%, 10.6%, 5.4% and 4.1%, respectively, when *x* = 0.05, 0.10, 0.15, 0.20, 0.30, 0.40, 0.50 and 0.60. These results suggest that the designed nanoparticles were appropriate for solid-state lighting.

Aside from quantum efficiency, thermal stability is another important parameter to characterize the performance of luminescent materials, as well as their possible feasibilities. To inspect the thermal quenching features of studied compounds, the emission spectra of KGd_2_F_7_:0.60Eu^3+^ nanoparticles at diverse temperatures were detected and presented in Figure 4d. As the temperature increased from 303 to 523 K, the luminescence profiles did not change, namely, the emission bands were not shifted at an elevated temperature, whereas the emission intensity declined gradually, owing to the thermal quenching effect. Notably, when the temperature was elevated to 423 K, the fluorescence intensity still kept 70% of its starting value at room temperature (303 K), suggesting that the designed nanoparticles possessed good thermal stability, which allowed them to be employed in the area of high-power solid-state lighting. Furthermore, on account of the recorded temperature-dependent emission spectra, the activation energy (∆E) was also calculated by using the following function [35,36]:(12)$$I=\frac{I_{0}}{1+\exp\left(-\Delta E/kT\right)}$$
here I_0_ pertains to the fluorescence intensity at initial temperature, I is associated with the fluorescence intensities at temperature *T*, and *k* represents Boltzmann constant. To find out the exact ΔE value, the plot of ln(I_0_/I−1) versus 1/*kT* was used, as shown in Figure 4f. As demonstrated, these data were able to be fitted by a straight line (the slope was −0.18), indicating that the ΔE value of Eu^3+^ in designed nanoparticles was 0.18 eV.

### 3.3. EL Performance of Fabricated White-LED

For the purpose of validating the viability of the developed nanoparticles for solid-state lighting application, a warm white-LED was designed via the utilization of a commercial NUV chip, which was bought from EPILEDS. Its central wavelength and full width at half maximum were 400 and 14.4 nm, respectively, KGd_2_F_7_:0.60Eu^3+^ nanoparticles, commercial BaMgAl_10_O_17_:Eu^2+^ (BAM:Eu^2+^) blue-emitting and (Ba,Sr)_2_SiO_4_:Eu^2+^ (BaSrSi:Eu^2+^) green-emitting phosphors. Then, the 0.2 g of KGd_2_F_7_:0.6Eu^3+^ nanoparticles, 0.09 g of BAM:Eu^2+^ and 0.06 g of BaSrSi:Eu^2+^ phosphors were weighed and mixed thoroughly in a translucent silicone epoxy (2 g). After that, the aforementioned mixture was coated onto the surface of a commercial NUV chip and heated at 100 °C for 2 h to obtain the white-LED. Figure 5a displays the EL emission spectrum of designed white-LED under the injection current of 100 mA. Clearly, the featured emission peaks of NUV chip, BAM:Eu^2+^, KGd_2_F_7_:0.60Eu^3+^ and BaSrSi:Eu^2+^ were detected in the measured EL emission profile. When the injection current was 100 mA, bright warm white light was seen in the designed device, as is illustrated in Figure 5b, in which its CIE coordinate, luminous efficiency, CCT and CRI were (0.362,0.359), 17.2 lm/W, 4456 K and 88.3, respectively. Furthermore, as the forward bias current rosed from 50 to 300 mA, the designed white-LED became hotter and hotter, namely, its temperatures increased (see Figure 5c). These achievements manifest that the EL performances of phosphor-converted white-LED are able to be modified by adopting Eu^3+^-activated KGd_2_F_7_ nanoparticles as red-emitting components.

## 4. Conclusions

In conclusion, the highly efficient KGd_2_F_7_:2*x*Eu^3+^ nanoparticles were prepared by an easy chemical precipitation technology at room temperature. The as-prepared samples not only possessed a pure monoclinic phase, but also exhibited homogeneous nanoparticles. Excited by 393 nm, all of the nanoparticles emitted dazzling red emissions and their intensities were dependent on Eu^3+^ content, in which the concentration quenching took place when *x* > 0.30. The electric dipole–dipole interaction brought about a concentration quenching effect and the *R_c_* value between Eu^3+^ in KGd_2_F_7_ host lattices was 6.29 Å. According to the Judd–Ofelt theory and emission spectrum, it is known that Eu^3+^ takes up high symmetry sites in KGd_2_F_7_ host lattices. Moreover, the IQE and EQE values of resultant nanoparticles were 95.5% and 32.9%, respectively, and excited at 393 nm. Furthermore, the prepared nanoparticles were also stable at a high temperature and the corresponding ∆E value was 0.18 eV. Ultimately, via the use of KGd_2_F_7_:0.60Eu^3+^ nanoparticles as red-emitting constituents, a high-quality white-LED was developed, in which its color coordinate, luminous efficiency, CCT and CRI were (0.362,0.359), 17.2 lm/W, 4456 K and 88.3, respectively, at the driven current of 100 mA. Thereby, all of the achievements suggest that Eu^3+^-activated KGd_2_F_7_ nanoparticles have the capacity to improve the quality of phosphor-converted white-LED through supplying the red-emitting components.

## Figures and Tables

**Figure 1 nanomaterials-12-04397-f001:**
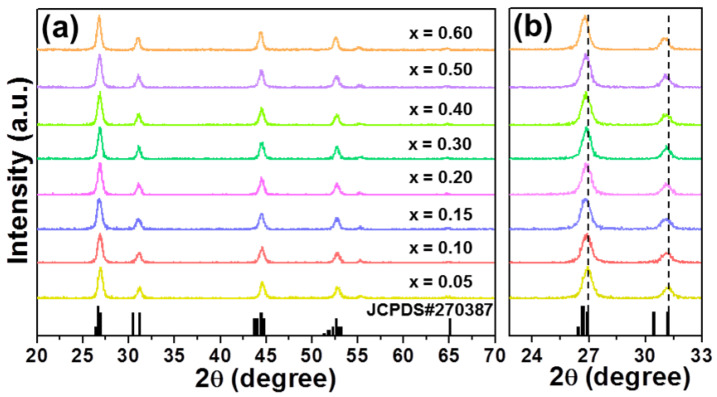
(**a**) XRD patterns of KGd_2_F_7_:2*x*Eu^3+^ (0.05 ≤ *x* ≤ 0.60) nanoparticles. (**b**) Zoomed XRD patterns of KGd_2_F_7_:2*x*Eu^3+^ (0.05 ≤ *x* ≤ 0.60) nanoparticles in the 2θ range of 23–33°.

**Figure 2 nanomaterials-12-04397-f002:**
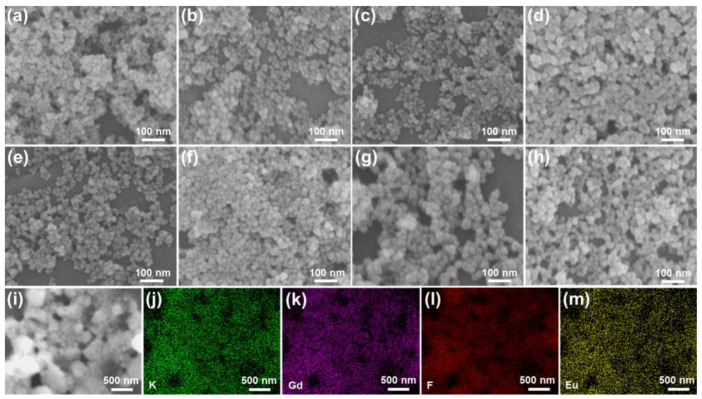
FE-SEM graphs of KGd_2_F_7_:2*x*Eu^3+^ nanoparticles with the doping contents of (**a**) *x* = 0.05, (**b**) *x* = 0.10, (**c**) *x* = 0.15, (**d**) *x* = 0.20, (**e**) *x* = 0.30, (**f**) *x* = 0.40, (**g**) *x* = 0.50 and (**h**) *x* = 0.60. (**i**–**m**) Elemental mapping of KGd_2_F_7_:0.60Eu^3+^ nanoparticles.

**Figure 3 nanomaterials-12-04397-f003:**
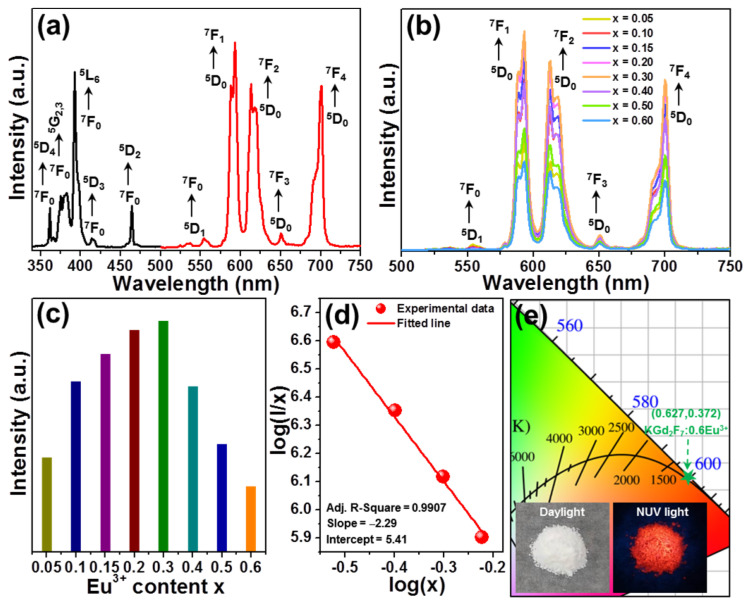
(**a**) Excitation and emission profiles of KGd_2_F_7_:0.60Eu^3+^ nanoparticles. (**b**) Emission profiles of KGd_2_F_7_:2*x*Eu^3+^ nanoparticles. (**c**) Dependence of fluorescence intensity on dopant concentration. (**d**) Plot of log(*x*) *vs.* log(*I*/*x*) for resultant nanoparticles. (**e**) CIE chromaticity diagram of KGd_2_F_7_:0.60Eu^3+^ nanoparticles as well as its optical images.

**Figure 4 nanomaterials-12-04397-f004:**
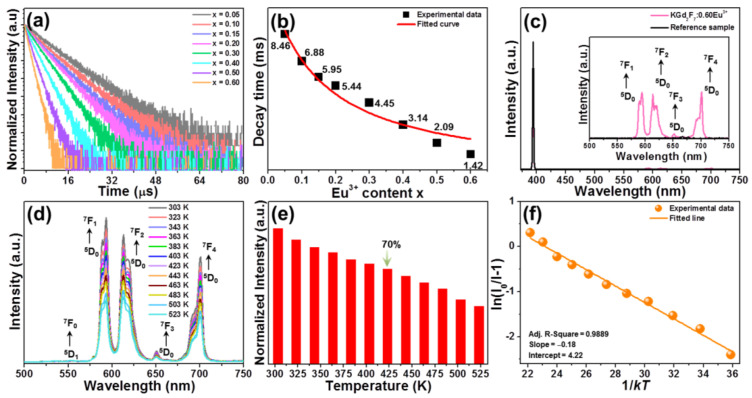
(**a**) Decay curves of KGd_2_F_7_:2*x*Eu^3+^ nanoparticles. (**b**) Lifetime of Eu^3+^ as a function of doping content. (**c**) Excitation profile of BaSO_4_ reference sample and emission profile of KGd_2_F_7_:0.60Eu^3+^ nanoparticles. (**d**) Emission spectra of KGd_2_F_7_:0.60Eu^3+^ nanoparticles as a function of temperature. (**e**) Normalized emission intensity at diverse temperatures. (**f**) Plot of ln(I_0_/I−1) *vs*. 1/*kT*.

**Figure 5 nanomaterials-12-04397-f005:**
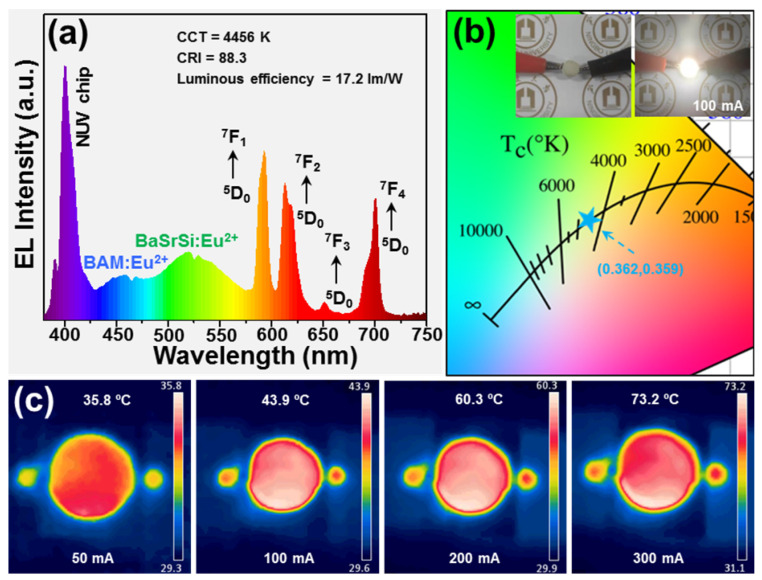
(**a**) EL emission profile and (**b**) CIE chromaticity diagram of packaged white-LED under the injection current of 100 mA. Inset of (**b**) displays the optical graphs of developed white-LED. (**c**) Thermal photos of packaged white-LED at various driven currents.

## Data Availability

All of the relevant data are available from the correspondence authors upon reasonable request. Source data are provided with this paper.

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
