# Peer review of "Room-Temperature Synthesis of Highly-Efficient Eu3+-Activated KGd2F7 Red-Emitting Nanoparticles for White Light-Emitting Diode"

_nanomaterials, 2022, doi:10.3390/nano12244397_

Round 1

Reviewer 1 Report

The authors report on the properties of a red emitting phosphor, with composition K(Gd,Eu)2F7. The structural and optical properties are discussed. The phosphor particles are applied to a near-UV LED in order to generate white light (in combination with other phosphors). The particles are nanosized, although this aspect is not of specific advantage in the envisioned application.

The manuscript is a rather basic report on the properties. Although most experiments seem well conducted and reported, there are a number of issues that require attention.

*) Regarding the structural characteristics of the particles, it is mentioned that “Moreover, the positions of diffraction peaks gradually shift to smaller angles caused by the different ionic sizes between doping ions (Eu3+) and replaced ions (Gd3+), as illustrated in Figure 1(b), further indicating that Eu3+ is incorporated into the KGd2F7 host lattices by occupying the Gd3+ site.”. This should be quantified, as the trend is not clear from the figure. For instance, the shift for x = 0.3 appears larger than the shift for x = 0.6.

*) The maps in figure 2(j)-(m) are too dark and should be visualized differently.

*) Figure 3b: the emission peak around 560nm is not related to the 5D0-7F0 transition. They originate from the 5D1 level, and are expected to decrease when the Eu concentration increases. This is in contrast to the statement that “their luminescence profiles are not determined by Eu3+ content”. Also in figure 4, the assignment is wrong.

*) The language should be checked. Some words contain typos (e.g. decied instead of decided, manifesting should be manifesting, ‘wroks” should be works, k\alpha should be K\alpha), some expression are not very scientific (‘dazzling blue emissions”), ‘pertians” should be pertains, Boltamann should be Boltzmann, ‘temperatrue” should be temperature…

*) regarding the type of energy transfer, the value of 6.87 is almost as close to 8 as to 6, so the statement “very close to 6” is not correct.

*) Figure 3e: the wavelength of the used NUV should be specified, as well as a mention of the use of any optical filter.

*) In equation (3), I0 is the intensity at times much longer than the decay time tau. It is not the intensity at time zero (which is I_0 + A, according to eq. (3)).

*) I don’t understand why the Judd-Ofelt analysis is performed on a sample where there is a strong impact of the dopant concentration, as witnessed by the strongly reduced lifetime. This compromises the results. Also in eq. (4),(5), J = 6 is missing, while J = 4 is mentioned twice.

*) Literature references should be added for the refractive index of compounds similar to K(Gd,Eu)2F7. Is a value of 1.92 in line with similar compounds?

*) I am not convinced that the J-O analysis is a real added value to this work, especially since more detailed spectroscopy works into KGd2F7:Eu have already been reported in literature (Journal of Alloys and Compounds 289 (1999) 71–80; and somewhat related: Journal of Luminescence 143 (2013) 293–297). The authors should compare their results to those studies.

*) The internal quantum efficiency is estimated at 95.5%. This is however very unlikely and needs further verification. There are two elements which make this unlikely: first of all, the decay time is already strongly reduced for the studied sample x = 0.3 compared to low doping concentrations (4.5ms compared to 8.5ms), which is an indication of non-radiative decay paths. Hence a QY for x = 0.3 of almost 100% is unlikely. Second, the EQE hardly increases when going from x = 0.15 to x = 0.30 (Figure 3c), whereas one expects the absorption to double. This means that the QY of x = 0.15 should be considerably higher than 100%, which is impossible. It would there for be very informative to add a graph with IQE, AE and EQE as a function of x. This could already reveal some measurements artefacts.

*) Several figures are missing values on the vertical axis.

*) Equation (9) is very strange, as for the AE one should look at the absorption of the excitation light (possibly via the amount of reflected light). This is not the same as an excitation spectrum. For E_R the reference is BaSO4. This material does not show luminescence, hence the excitation spectrum E_R will also be zero. When integrating, this would lead to the denominator of eq. 9 being zero, which does not make sense.

*) On line 271 it is mentioned “that the activation energy was calculated in order to deeply reveal the thermal quenching effect”. I don’t see why deriving a value of 0.18eV ‘would deeply reveal the thermal quenching effect’.

*) A white LED was constructed. The specifications and the used amounts of BAM:Eu, BaSrSi(?):Eu and the KGd2F7:Eu phosphor should be mentioned (also, the text mentions CaSrSi on line 291, where other compositions are mentioned above.

*) The overall efficiency for a white LED is very low (17 lum/W). What was the electrical to optical conversion efficiency of the pumping LED? Also more details on this LED should be provided (FWHM, peak wavelength). The emission spectrum of this LED, without phosphors, should be added to figure 5a.

*) for high power applications, the light output as a function of optical input power should be studied, preferably for the K(Gd,Eu)2F7 phosphor only on top of the NUV LED (so without the other phosphors) or upon laser excitation with variable power density.

Author Response

Dear editor, dear reviewers:

We are pleased to get that our manuscript has received a positive review. Thank you and the reviewers for reviewing my manuscript and pointing out the significant objections. We think we have complied with the reviewers' comments in our revised manuscript. We hope our manuscript could be accepted this time.

Responding to the comments of Reviewer #1:

  1. Regarding the structural characteristics of the particles, it is mentioned that “Moreover, the positions of diffraction peaks gradually shift to smaller angles caused by the different ionic sizes between doping ions (Eu3+) and replaced ions (Gd3+), as illustrated in Figure 1(b), further indicating that Eu3+ is incorporated into the KGd2F7 host lattices by occupying the Gd3+ site.”. This should be quantified, as the trend is not clear from the figure. For instance, the shift for x = 0.3 appears larger than the shift for x = 0.6.

Responding: Thank you very much for pointing out this. We measured the XRD patterns of KGd2F7:2xEu3+ nanoparticles with the x value of 0.30 and 0.60 again, as shown in Figure 1. Clearly, the KGd2F7:2xEu3+ (x = 0.60) nanoparticles exhibit larger shift than the KGd2F7:2xEu3+ (x = 0.30) nanoparticles. Moreover, it can be seen that the diffraction bands shift to smaller angles with the increment of Eu3+ content. Herein, the shift phenomenon is not serious, which is caused by the smaller difference between the ionic radii of Eu3+ (i.e., 0.947 Å) and Gd3+ (i.e., 0.938 Å).

  1. The maps in figure 2(j)-(m) are too dark and should be visualized differently.

Responding: Thank you very much for pointing out this. The elemental mapping results have been measured again and the corresponding results are shown in the revised version, in which the quality of figure has been improved.

  1. Figure 3b: the emission peak around 560 nm is not related to the 5D0-7F0 transition. They originate from the 5D1 level, and are expected to decrease when the Eu concentration increases. This is in contrast to the statement that “their luminescence profiles are not determined by Eu3+ content”. Also in figure 4, the assignment is wrong.

Responding: Thank you very much for pointing out this. The aforementioned mistakes are modified in the revised manuscript. According to the recorded the emission spectra (see Figure 3(b)), one knows that the positions of the emission bands are changed with the increment of Eu3+ content. Thus, we stated that the luminescence profiles are not determined by Eu3+ content. For the sake of eliminating the misunderstanding, the “their luminescence profiles are not determined by Eu3+ content” is removed from the manuscript. Furthermore, the “As disclosed in Figure 3(a), when the excitation wavelength is 393 nm, there are five sharp peaks in the recorded luminescence spectrum, in which their central wavelengths are 554, 593, 613, 650 and 700 nm corresponding to the intra-4f transitions of Eu3+, that is, 5D17F0, 5D07F1, 5D07F2, 5D07F3 and 5D07F0, respectively [23,24].” was supplied in the revised manuscript (Page 4 paragraph 2 line 10-13).

  1. The language should be checked. Some words contain typos (e.g. decied instead of decided, manifesting should be manifesting, ‘wroks” should be works, k\alpha should be K\alpha), some expression are not very scientific (‘dazzling blue emissions”), ‘pertians” should be pertains, Boltamann should be Boltzmann, ‘temperatrue” should be temperature…

Responding: Thank you very much for pointing out these. These aforementioned mistakes have been corrected in the revised manuscript. Furthermore, we also checked the manuscript carefully and the English expression is improved in the revised version.

  1. Regarding the type of energy transfer, the value of 6.87 is almost as close to 8 as to 6, so the statement “very close to 6” is not correct.

Responding: Thank you very much for pointing out this. To study the concentration quenching mechanism of rare-earth ions doped phosphors, the expression of  is widely used to analyze the relation between emission intensity and doping content so as to obtain the θ value. As for θ, it has three different values of 6, 8 and 10 corresponding to dipole-dipole, dipole-quadrupole and quadrupole-quadrupole interactions, respectively (J. Lumin. 225 (2020) 117365; J. Lumin. 221 (2020) 117105; J. Am. Ceram. Soc. 103 (2020) 1037). Herein, we also used the aforementioned function to investigate the concentration quenching mechanism of designed nanoparticles and found that θ value was 6.87. Note that, the difference between calculated θ value and 6 is 0.87, while the difference between the calculated θ value and 8 is 1.13. Significantly, compared with 8, the obtained θ value is more close to 6. Similar phenomenon was also reported in other luminescent materials, such as La2Mo2O9:Eu3+ (θ = 6.18; J. Alloys Compd. 783 (2019) 969), Ca2MgTeO6:Sm3+ (θ = 6.33; J. Lumin. 225 (2020) 117374), La6Ba4Si6O24F2:Sm3+ (θ = 6.03; J. Lumin. 206 (2019) 417), Ca2LaTaO6:Mn4+ (θ = 6.06; J. Alloys Compd. 780 (2019) 749) and Ca2Gd8Si6O26:Eu3+ (θ = 6.03; J. Alloys Compd. 859 (2021) 157843), etc. Thereby, it is reasonable for us to consider that the concentration quenching mechanism in Eu3+-activated KGd2F7 nanoparticles is dominated by electric dipole-dipole interaction.

  1. Figure 3e: the wavelength of the used NUV should be specified, as well as a mention of the use of any optical filter.

Responding: Thank you very much for pointing out this. In present work, the wavelengths of NUV light and commercial NUV chip are all around 400 nm. Furthermore, the optical filter was used only in measuring the excitation and emission spectra. Specially, to record the excitation spectrum, an optical filter with the cutoff wavelength of 510 nm (i.e., λ ≤ 510 nm) was used, while another type of optical filter with the cutoff wavelength of 400 nm (λ ≥ 400 nm) was employed to measure the emission spectrum. Furthermore, the “Herein, to measure the excitation spectrum, an optical filter with the cutoff wavelength of 510 nm (i.e., λ ≤ 510 nm) was adopted, whereas a cutoff filter (λ ≥ 400 nm) was employed to recorded the emission spectrum.” was added in the revised manuscript (Page 3 paragraph 1 line 7-9).

  1. In equation (3), I0 is the intensity at times much longer than the decay time tau. It is not the intensity at time zero (which is I0 + A, according to eq. (3)).

Responding: Thank you very much for pointing out this. It has been corrected in the manuscript and “I0 is the fluorescence intensity at time t, which is much longer than the decay time τ” was supplied in the revised manuscript (Page 6 paragraph 2 line 6-7).

  1. I don’t understand why the Judd-Ofelt analysis is performed on a sample where there is a strong impact of the dopant concentration, as witnessed by the strongly reduced lifetime. This compromises the results. Also in eq. (4),(5), J = 6 is missing, while J = 4 is mentioned twice.

Responding: Thank you very much for pointing out this. According to previous literatures (J. Appl. Phys. 109 (2011) 053511; Ceram. Int. 42 (2016) 13648), one knows that the site properties of Eu3+ in luminescent materials can be analyzed by using the Judd-Ofelt theory based on the decay time and emission spectra. Herein, the Eq. (4)-(7) are used to calculate the optical transition parameters of Eu3+ in KGd2F4 host lattices. According to these functions, one knows that the Ωλ values are decided by the positions of the emission bands rather than their intensities. Form the recorded emission spectra (see Figure 3(b)), it is clear that positions of the emission peaks are hardly changed with the increment of Eu3+ content, which means that the Eu3+ content does not have any impacts on the Ωλ values of Eu3+ in resultant nanoparticles. Therefore, we chose the KGd2F7:2xEu3+ nanoparticles with the doping content of 30 mol% as the representative sample to study the symmetric properties of Eu3+ in designed compounds. In addition, the J value, which is shown in Eq. (4) and (5), is assigned to 0, 1, 2, 3, 4 and 6. Ultimately, the “J = 0, 1, 2, 3, 4, 4” has been modified as “J = 0, 1, 2, 3, 4, 6” in the revised manuscript.

  1. Literature references should be added for the refractive index of compounds similar to K(Gd,Eu)2F7. Is a value of 1.92 in line with similar compounds?

Responding: Thank you very much for pointing out this. To the best of our knowledge, there are no any reports on the refractive index of Eu3+-activated KGd2F7 nanoparticles. Thus, it is difficult for us to supply the corresponding references, which deal with the refractive index of Eu3+-activated KGd2F7 nanoparticles, at the moment.  In present work, we used Eq. (6), which had been also employed in previous works (CrystEngComm 14 (2012) 1760; J. Mater. Sci. 52 (2017) 935; Ceram. Int. 48 (2022) 15165), to calculate the refractive index of designed nanoparticles. Herein, the refractive index of KGd2F7:0.6Eu3+ nanoparticles is found to be 1.92.

10 I am not convinced that the J-O analysis is a real added value to this work, especially since more detailed spectroscopy works into KGd2F7:Eu have already been reported in literature (Journal of Alloys and Compounds 289 (1999) 71–80; and somewhat related: Journal of Luminescence 143 (2013) 293–297). The authors should compare their results to those studies.

Responding: Thank you very much for pointing out this. The Eu3+-activated KGd2F7 compounds, which were reported in previous literatures (J. Alloys Compd. 28 (1999) 71; J. Lumin. 143 (2013) 293), were all prepared by sintering at a high temperature. As for the Eu3+-activated KGd2F7 compounds reported by Pierrard et al. (J. Alloys Compd. 289 (1999) 71), they studied the influence of oxygen on the crystal structure and luminescence properties of resultant samples, in which the prepared samples with pure phase was obtained when the sintering temperature was 630 ºC, while the impurity phase occurred when the sintering temperature was increased to 850 ºC (see their XRD results). Note that, they did not present the emission spectrum of the studied samples sintered at 630 ºC and these emission spectra shown in the reference were all related to the final products prepared at 850 ºC (J. Alloys Compd. 289 (1999) 71), in which the intensity of the emission originating from 5D07F1 transition was stronger than that of the emission from 5D07F1 transition, suggesting that Eu3+ occupied the high symmetry sites in the studied samples. On the other hand, the Eu3+-activated KGd2F7 phosphors proposed by R. Lisiecki. (J. Lumin. 143 (2013) 293), which was prepared by solid-state reaction method, also emitted the featured emissions of Eu3+, where the intensity of the emission originating from 5D07F1 transition was also stronger than that of the emission from 5D07F1 transition. Evidently, these two previous works all confirmed that Eu3+ took place the high symmetry sites in KGd2F7 host lattices, which coincides well with our result. Note that, these two previous works only roughly reported the luminescence properties of Eu3+-activated KGd2F7 phosphors. In comparison, we used a new facile method to prepare the Eu3+-activated KGd2F7 nanoparticles at room temperature, which has never been reported before. Furthermore, we also systematically studied the phase structure, morphology, luminescence properties, decay time, quantum efficiency and thermal stability of Eu3+-activated KGd2F7 nanoparticles. Ultimately, via the use of the designed nanoparticles as red-emitting components, a warm white-LED was packaged so as to confirm its applications in solid-state lighting. Thereby, our present work is different from previous reports.

  1. The internal quantum efficiency is estimated at 95.5%. This is however very unlikely and needs further verification. There are two elements which make this unlikely: first of all, the decay time is already strongly reduced for the studied sample x = 0.3 compared to low doping concentrations (4.5ms compared to 8.5ms), which is an indication of non-radiative decay paths. Hence a QY for x = 0.3 of almost 100% is unlikely. Second, the EQE hardly increases when going from x = 0.15 to x = 0.30 (Figure 3c), whereas one expects the absorption to double. This means that the QY of x = 0.15 should be considerably higher than 100%, which is impossible. It would there for be very informative to add a graph with IQE, AE and EQE as a function of x. This could already reveal some measurements artefacts.

Responding: Thank you very much for pointing out this. To further confirm the luminescence properties of resultant nanoparticles, we sent our prepared nanoparticles to another testing organization to measure their quantum efficiencies as a function of Eu3+ content. According to the measured results, one knows that the IQE values of KGd2F7:2xEu3+ nanoparticles are 42.6%, 58.0%, 73.6%, 78.1%, 78.4%, 59.5%, 43.6% and 20.2%, respectively, when x = 0.05, 0.10, 0.15, 0.20, 0.25, 0.30, 0.40, 0.50 and 0.60. Furthermore, the EQE values of KGd2F7:2xEu3+ nanoparticles are found to be 6.0%, 8.3%, 9.4%, 13.4%, 16.1%, 10.6%, 5.4% and 4.1%, respectively, when x = 0.05, 0.10, 0.15, 0.20, 0.25, 0.30, 0.40, 0.50 and 0.60. Evidently, the IQE and EQE values of resultant nanoparticles are dependent on Eu3+ content. Similar changing tendency was also reported in other luminescent materials, such as La2CaSnO6:Eu3+ (J. Rare Earths. 40 (2022) 1682), La2Si2O7:Ce3+/Tb3+/Eu3+ (J. Alloys Compd. 785 (2019) 53), LiLaMgWO6:Eu3+ (J. Alloys Compd. 43 (2017) 2720), Ca2YSbO6:Eu3+ (RSC Adv. 9 (2019) 20742), and so forth. Compared with our previous measured results, these obtained data are not the same. This is because that the integrating sphere, which was initially used to measure the quantum efficiencies of resultant nanoparticles, has been contaminated, leading to the incorrect data. Additionally, the “Excited by 393 nm, the IQE values of KGd2F7:2xEu3+ nanoparticles are determined to be 42.6%, 58.0%, 73.6%, 78.1%, 59.5%, 43.6% and 20.2%, respectively, when x = 0.05, 0.10, 0.15, 0.20, 0.30, 0.40, 0.50 and 0.60, whereas their EQE values are 6.0%, 8.3%, 9.4%, 13.4%, 16.1%, 10.6%, 5.4% and 4.1%, respectively, when x = 0.05, 0.10, 0.15, 0.20, 0.30, 0.40, 0.50 and 0.60. These results suggest that the designed nanoparticles are appropriate for solid-state lighting.” was added in the revised manuscript (Page 8 paragraph 2 line 12-17).

  1. Several figures are missing values on the vertical axis.

Responding: Thank you very much for pointing out this. The units of the experimental data, which are shown in present work, are dimensionless, namely, arbitrary unit. Furthermore, the experimental data, such as XRD, excitation and emission spectra, etc., were all recorded at the same condition. Thereby, there is no need to show the values of y axis. Similar operation was also extensively carried out in previous literatures (J. Am. Chem. Soc. 140 (2018) 9730; Adv. Opt. Mater. 10 (2022) 2102287; Inorg. Chem. Front. 9 (2022) 3224; Inorg. Chem. Front. 9 (2022) 1644).

  1. Equation (9) is very strange, as for the AE one should look at the absorption of the excitation light (possibly via the amount of reflected light). This is not the same as an excitation spectrum. For ER the reference is BaSO4. This material does not show luminescence, hence the excitation spectrum ER will also be zero. When integrating, this would lead to the denominator of eq. 9 being zero, which does not make sense.

Responding: Thank you very much for pointing out this. In present work, via the use of Eqs. (8)-(10) as well as the recorded luminescence spectra, the IQE and EQE values of studied nanoparticles were obtained. Herein, AE refers to the absorption of the excitation light, whereas ER and ES are assigned to the integrated intensities of excitation light without and with developed nanoparticles in integrating sphere (J. Lumin. 232 (2021) 117857; J. Alloys Compd. 812 (2020) 152119; Chem. Eng. J. 405 (2021) 126950; Mater. Today. Eng. 17 (2020) 100448). Furthermore, the “AE is the absorption of the excitation light, LS stands for the emission spectrum of designed compounds, ES and ER are assigned to the integrated intensities of excitation light with and without developed nanoparticles in integrating sphere, respectively.” was added in the revised manuscript (Page 8 paragraph 1 line 8-11).

  1. On line 271 it is mentioned “that the activation energy was calculated in order to deeply reveal the thermal quenching effect”. I don’t see why deriving a value of 0.18eV ‘would deeply reveal the thermal quenching effect’.

Figure R1. Schematic configuration coordinate diagram of the thermal quenching process in Eu3+-activated KGd2F7 nanoparticles.

Responding: Thank you very much for pointing out this. For the sake of describing the luminescence process of Eu3+-activated phosphors, the schematic configuration coordinate diagram of Eu3+ was plotted and shown in Figure R1. According to previous literature (Ceram. Int. 44 (2018) 1909), one knows that there is a crossing point (i.e., B) between 7FJ and 5D0 levels, as shown in Figure R1. Excited by 393 nm, electrons can be excited from 7F0 to 5L6 levels, and then nonradiative transition takes place, leading to the population of 5D0 level. When the surrounding temperature is low, electrons will return to the 7FJ (J = 0-6) ground state from the minimum energy point A of 5D0 level, resulting in the featured emissions of Eu3+ (see Figure R1). However, with the increment of temperature, parts of electrons located at A point can overcome the energy barrier ∆E (i.e., activation energy) to reach B point, and then nonradiatively decay to the 7FJ ground state via pathway 1, as shown in Figure R1, realizing the quenched emission intensity at elevated temperature. Clearly, the activation energy plays an important role in determining the thermal quenching performance of luminescent materials, namely, larger activation energy leads to higher thermal stability. Thereby, we estimated the activation energy of Eu3+ in KGd2F7 nanoparticles. Furthermore, to eliminate the misunderstanding, we have modified the expression and the “Furthermore, on account of the recorded temperature-dependent emission spectra, the activation energy (∆E) was also calculated by using the following function” was added in the revised manuscript (Page 9 paragraph 1 line 1-3).

  1. A white LED was constructed. The specifications and the used amounts of BAM:Eu, BaSrSi:Eu and the KGd2F7:Eu phosphor should be mentioned (also, the text mentions CaSrSi on line 291, where other compositions are mentioned above.

Responding: Thank you very much for pointing out this. To fabricate the white-LED, the designed KGd2F7:0.6Eu3+ nanoparticles, commercial BAM:Eu2+ and BaSrSi:Eu2+ phosphors were used. Herein, 0.2 g of KGd2F7:0.6Eu3+ nanoparticles, 0.09 g of BAM:Eu2+ and 0.06 g of BaSrSi:Eu2+ phosphors were weighted and thoroughly mixed in translucent silicone epoxy (2 g). After that, the above mixture was coated the surface of a commercial NUV (400 nm) chip and heated at 100 ºC for 2 h to obtain the white-LED. Thereby, the masses of KGd2F7:0.6Eu3+ nanoparticles, commercial BAM:Eu2+ and BaSrSi:Eu2+ phosphors were 0.2, 0.09 and 0.06 g, respectively. Furthermore, the “Herein, the 0.2 g of KGd2F7:0.6Eu3+ nanoparticles, 0.09 g of BAM:Eu2+ and 0.06 g of BaSrSi:Eu2+ phosphors were weighted and mixed thoroughly in translucent silicone epoxy (2 g). After that, the aforementioned mixture was coated on the surface of a commerical NUV chip and heated at 100 ºC for 2 h to obtian the white-LED.” was added in the revised manuscript (Page 9 paragraph 2 line 5-9). In addition, the “CaSrSi” should be “BaSrSi:Eu2+” and it has been corrected in the revised manuscript.

  1. The overall efficiency for a white LED is very low (17 lm/W). What was the electrical to optical conversion efficiency of the pumping LED? Also more details on this LED should be provided (FWHM, peak wavelength). The emission spectrum of this LED, without phosphors, should be added to figure 5a.

Responding: Thank you very much for pointing out this. In present work, we prepared a NUV chip based warm white-LED. As is known, the intrinsic efficiency of the commercial NUV chip is low which largely restrict the luminous efficiency of fabricated white-LED. Furthermore, the central wavelength of commercial NUV chip is not the same as the excitation wavelengths of the designed nanoparticles and commercial phosphors, implying that the luminescent materials are not able to be efficiently excited by the commercial NUV chip. Owing to these characteristics, the NUV chip based white-LED usually exhibits relatively low luminous efficiency. Similar results were also reported in other NUV chip pumped white-LED (Chem. Eng. J. 405 (2021) 126950 (27.4 lm/W); J. Rare Earths. 39 (2021) 1040 (10.5 lm/W); Dyes Pigments 157 (2018) 40 (1.217 lm/W); Inorg. Chem. 61 (2022) 6898 (7.58 lm/W)).

On the other hand, due to the limitation of our device, it is difficult for us to measure the optical conversion efficiency of the pumping LED at the moment. Furthermore, the central wavelength and full width at half maximum (FWHM) are 400 and 14.4 nm, respectively. In present work, we tried to explore the promising application of designed nanoparticles for solid-state lighting application, in which the commercial NUV chip acts as the excitation lighting source. Note that, the luminescence performance of commercial NUV chip is not our research target. Thereby, it is meaningless for us to investigate the emission spectrum of commercial NUV chip. In addition, the emission band of the commercial NUV chip can also be observed in the recorded EL emission spectrum of packaged white-LED, as illustrated in Figure 5(a). Ultimately, the “a warm white-LED was designed via the utilization of a commercial NUV chip, which was bought from EPILEDS and its central wavelength and full width at half maximum are 400 and 14.4 nm, respectively,” was added in the revised manuscript (Page 9 paragraph 2 line 2-4).

  1. for high power applications, the light output as a function of optical input power should be studied, preferably for the K(Gd,Eu)2F7 phosphor only on top of the NUV LED (so without the other phosphors) or upon laser excitation with variable power density.

Figure R2. EL emission spectra of packaged white-LED as a function of injection current.

Table R1. CCT, CRI, color coordinates and luminous efficiency of white-LED as a function of driving current in the range of 50-300 mA. 

Current

CIE coordinate

CCT

CRI

Luminous efficiency

50 mA

(0.366,0.363)

4310 K

88.0

16.8 lm/W

100 mA

(0.362,0.359)

4456 K

88.3

17.2 lm/W

200 mA

(0.360,0.357)

4512 K

88.3

17.3 lm/W

300 mA

(0.357,0.356)

4593 K

88.5

16.9 lm/W

Responding: Thank you very much for pointing out this. To check the impact of injection current on the EL properties of packaged white-LED, its EL emission spectra as a function of injection current were measured, as displayed in Figure R2. As disclosed, the emission bands are not shifted at high injection current, whereas their intensities are increased gradually with raising the current from 50 to 300 mA. Moreover, the EL properties, such as CCT, CRI, color coordinate and luminous efficiency, of developed white-LED are also impacted by the driving current, as listed in Table R1. These results suggest that the packaged white-LED has stable EL properties at high injection current. Nevertheless, the main highlights of our present work are utilizing a facile method to prepare the Eu3+-activated KGd2F7 nanoparticles at room temperature and reveal their luminescence properties, and finally explore their promising applications in solid-state lighting. In present work, we did not try to confirm that the designed nanoparticles can be used for high power solid-state lighting application. Therefore, it is no need for us to supply the EL behaviors of developed white-LED at diverse forward bias currents at the moment. On the other hand, based on the recorded EL emission spectra (see Figure R2), it is clear that the designed nanoparticles can be excited by the commercial NUV chip, in which the emission intensities of the resultant nanoparticles are enhanced with increasing the injection current. Thereby, we did not supply the EL data of the LED, which was composed of the developed KGd2F7:0.6Eu3+ nanoparticles and a commercial NUV chip, in the manuscript.

Reviewer 2 Report

The paper deal with synthesis and spectroscopic properties of KGd2F7 doped with Eu3+. To consider this manuscript for publication following issues should be taken into account:

-          The manufacturer and purity of chemicals used should be added

-          “stirring” instead “agitation”

-          “When the reaction process was finished (stirring for 2 h)…” how the authors stated that the synthesis is finished after 2 h?

-          Authors wrote “diffraction peaks gradually shift to smaller angles… as illustrated in Figure 1(b)”. I don’t see it. Maybe it would help if authors will add vertical line crossed the peak.

-          Authors wrote “…sizes are around 23 nm...” How the size of the particles was evaluated?

-          The scale in figure 2(a)-2(h)is unreadable

-          The contrast in the Fig 2(j)-2(m) should be increased as it is difficult to see anything.

-          Line 124 “…typical KGd2F7:0.60Eu3+” now is the question what is the composition of the compound? KGd1.4Eu0.6F7 or KGd1.7Eu0.3F7?

-          Line 156 “and their luminescence profiles are not determined byEu3+ content.” I’m not sure what authors wanted to write here? That the positions of the peaks remain, the FWHM of peaks are the same, intensity ratio of the peaks are constant?

-          “where Z denotes the quantity of cation sites in the studied samples and V refers to the 167volume of unit cell. In this work, the values of xc, Z and V for synthesized nanoparticles 168are 0.30, 2 and 78.11 Å3, respectively.” As the authors just briefly describe structural part of the work (nothing about unit cell parameters) authors should refer from where they taken these values.

-          Line 193 “luminescence kinetic” instead of “luminescent dynamic”

-          The lifetime should be denoted in ms instead of ms

-          In luminescence part authors wrote “When Eu3+content is over 30 mol%, concentration quenching occurs” and in kinetic part is written and shown that the concentration quenching occurs already at low concentrations. What is the reason of these differences?

-          What is the QE and EQE for other Eu concentrations?

-          “commercial  NUV  chip” Manufacturer, type?

Author Response

Dear editor, dear reviewers:

We are pleased to get that our manuscript has received a positive review. Thank you and the reviewers for reviewing my manuscript and pointing out the significant objections. We think we have complied with the reviewers' comments in our revised manuscript. We hope our manuscript could be accepted this time.

Responding to the comments of Reviewer #2:

  1. The manufacturer and purity of chemicals used should be added

Responding: Thank you very much for pointing out this. In present work, all of the chemicals were bought from Aladdin Company. Furthermore, the purities of KNO3, Gd(NO3)3·6H2O, Eu(NO3)3·6H2O and NH4F are 99%, 99.9%, 99.99% and 98%, respectively. In addition, the “The raw materials were KNO3, Gd(NO3)3·6H2O, Eu(NO3)3·6H2O and NH4F, which were all bought from Aladdin Company and their purities were 99%, 99.9%, 99.99% and 98%, respectively.” was added in the revised manuscript (Page 2 paragraph 3 line 3-5).

  1. “stirring” instead “agitation”

Responding: Thank you very much for pointing out this. It has been modified in the revised version.

  1. “When the reaction process was finished (stirring for 2 h)…” how the authors stated that the synthesis is finished after 2 h?

Responding: Thank you very much for pointing out this. In present work, to prepare the Eu3+-activated KGd2F7 nanoparticles, we used a facile chemical precipitation reaction method, in which the reaction time was 2 h. As for the statement of the “when the reaction process was finished (stirring 2 h)…”, it means that the mixture was stirred for 2 h at room temperature. After stirring for 2 h, the obtained mixture was centrifuged, washed and heated to obtain the final products. Furthermore, to eliminate misunderstanding, the “when the reaction process was finished (stirring 2 h)…” was changed as “After stirring for 2 h” in the revised manuscript (Page 2 paragraph 3 line 10).

  1. Authors wrote “diffraction peaks gradually shift to smaller angles… as illustrated in Figure 1(b)”. I don’t see it. Maybe it would help if authors will add vertical line crossed the peak.

Responding: Thank you very much for pointing out this. We have modified Figure 1(b) and a vertical line was supplied in Figure 1(b).

  1. Authors wrote “…sizes are around 23 nm...” How the size of the particles was evaluated?

Responding: Thank you very much for pointing out this. In present work, we used a software, namely, Nano Measurer 1.2, to calculate the particle size of the resultant nanoparticles based on the recorded FE-SEM images.

  1. The scale in figure 2(a)-2(h) is unreadable

Responding: Thank you very much for pointing out this. The quality of Figure 2(a)-2(h) has been modified and the corresponding scale was also enlarged.

  1. The contrast in the Fig 2(j)-2(m) should be increased as it is difficult to see anything.

Responding: Thank you very much for pointing out this. The elemental mapping results have been measured again and their quality have been improved.

  1. Line 124 “…typical KGd2F7:0.60Eu3+” now is the question what is the composition of the compound? KGd1.4Eu0.6F7 or KGd1.7Eu0.3F7?

Responding: Thank you very much for pointing out this. In present work, the chemical formula of final product is KGd2-2xF7:2xEu3+ and it is short as KGd2F7:2xEu3+. Therefore, the KGd2F7:0.60Eu3+ stands for KGd1.4F7:0.60Eu3+ in present work.  

  1. Line 156 “and their luminescence profiles are not determined by Eu3+ content.” I’m not sure what authors wanted to write here? That the positions of the peaks remain, the FWHM of peaks are the same, intensity ratio of the peaks are constant?

Responding: Thank you very much for pointing out this. In present work, we wanted to use the sentence of the “and their luminescence profiles are not determined by Eu3+ content” to state that the positions of the emission bands are not changed by increasing the doping content. From the normalized emission spectra (see Figure R1), it is clear that the luminescence profiles are almost the same, in which their band positions, FWHEM values of emission bands and intensity ratio of emission peaks are nearly the same. Thereby, in previous version, we stated that the luminescence profiles are not determined by Eu3+ content. Herein, for the sake of eliminating misunderstanding, we modified the sentence and the “their positions are hardly changed by altering Eu3+ content.” was added in the revised manuscript (Page 5 paragraph 2 line 5-6).

Figure R1. Normalized emission spectra of Eu3+-activated KGd2F7 nanoparticles as a function of Eu3+ content.

  1. “where Z denotes the quantity of cation sites in the studied samples and V refers to the 167 volume of unit cell. In this work, the values of xc, Z and V for synthesized nanoparticles 168 are 0.30, 2 and 78.11 Å3, respectively.” As the authors just briefly describe structural part of the work (nothing about unit cell parameters) authors should refer from where they taken these values.

Responding: Thank you very much for pointing out this. In present work, the phase structure of studied samples was characterized by X-ray diffraction (XRD). According to the XRD results (Figure 1(a)), one knows that the recorded XRD patterns are able to be indexed by the standard KGd2F7 (JCPDS#270387), implying that the final products have pure monoclinic phase. From previous report (Cryst. Growth Des. 19 (2019) 2340), one knows that the lattice parameters of KGd2F7 host lattices are: a = 4.039 Å, b = 4.045 Å, c = 5.852 Å, α = γ = 90º, β = 107.92 º and Z = 2. Thus, the “the values of xc, Z and V for synthesized nanoparticles are 0.30, 2 and 78.11 Å3, respectively.” was presented in the manuscript. Furthermore, the corresponding reference, which was marked as 18, was supplied in the revised manuscript.

  1. Line 193 “luminescence kinetic” instead of “luminescent dynamic”

Responding: Thank you very much for pointing out this. It has been corrected in the revised manuscript.

  1. The lifetime should be denoted in ms instead of ms

Responding: Thank you very much for pointing out this. The unit of lifetime has been modified as ms in the revised version.

  1. In luminescence part authors wrote “When Eu3+ content is over 30 mol%, concentration quenching occurs” and in kinetic part is written and shown that the concentration quenching occurs already at low concentrations. What is the reason of these differences?

Responding: Thank you very much for pointing out this. From previous literatures (J. Am. Ceram. Soc. 102 (2019) 5910; J. Lumin. 181 (2017) 332; J. Lumin. 212 (2019) 23), one knows that the experimental decay time (τ) of involved emission is composed of radiative and non-radiative decay from the same state, in which their relation can be expressed as: τ = 1/(kr + ki) (kr and ki are assigned to the probabilities of radiative decay and non-radiative decay processes, respectively). With the increment of doping content, kr or ki is possible to be increased, leading to declined lifetime at elevated dopant content. Herein, when the doping content is not high enough (i.e., x ≤ 0.30), the probability of radiative decay process will be increased, resulting in improved emission intensity. In comparison, the probability of nonradiative decay will take the domination when the Eu3+ content is over 30 mol%, leading to the concentration quenching effect as well as the declined emission intensity. Similar phenomenon was also intensively reported in other luminescent materials, such as Ca3Y2(Si3O9)2:Dy3+ (Dalton Trans. 43 (2014) 11474), Na3AlF6:Cr3+ (ACS Appl. Electron. Mater. 1 (2019) 2325); Na2ZnSiO4:Mn2+ (J. Lumin. 213 (2019) 1); Sr2YSbO6:Mn4+ (Chem. Eng. J. 412 (2021) 128633), etc.

  1. What is the QE and EQE for other Eu concentrations?

Responding: Thank you very much for pointing out this. To further confirm the luminescence properties of resultant nanoparticles, we sent our prepared nanoparticles to another testing organization to measure their quantum efficiencies as a function of Eu3+ content. According to the measured results, one knows that the IQE values of KGd2F7:2xEu3+ nanoparticles are 42.6%, 58.0%, 73.6%, 78.1%, 78.4%, 59.5%, 43.6% and 20.2%, respectively, when x = 0.05, 0.10, 0.15, 0.20, 0.25, 0.30, 0.40, 0.50 and 0.60. Furthermore, the EQE values of KGd2F7:2xEu3+ nanoparticles are found to be 6.0%, 8.3%, 9.4%, 13.4%, 16.1%, 10.6%, 5.4% and 4.1%, respectively, when x = 0.05, 0.10, 0.15, 0.20, 0.25, 0.30, 0.40, 0.50 and 0.60. Evidently, the IQE and EQE values of resultant nanoparticles are dependent on Eu3+ content. Similar changing tendency was also reported in other luminescent materials, such as La2CaSnO6:Eu3+ (J. Rare Earths. 40 (2022) 1682), La2Si2O7:Ce3+/Tb3+/Eu3+ (J. Alloys Compd. 785 (2019) 53), LiLaMgWO6:Eu3+ (J. Alloys Compd. 43 (2017) 2720), Ca2YSbO6:Eu3+ (RSC Adv. 9 (2019) 20742), and so forth. Compared with our previous measured results, these obtained data are not the same. This is because that the integrating sphere, which was initially used to measure the quantum efficiencies of resultant nanoparticles, has been contaminated, leading to the incorrect data. Additionally, the “Excited by 393 nm, the IQE values of KGd2F7:2xEu3+ nanoparticles are determined to be 42.6%, 58.0%, 73.6%, 78.1%, 59.5%, 43.6% and 20.2%, respectively, when x = 0.05, 0.10, 0.15, 0.20, 0.30, 0.40, 0.50 and 0.60, whereas their EQE values are 6.0%, 8.3%, 9.4%, 13.4%, 16.1%, 10.6%, 5.4% and 4.1%, respectively, when x = 0.05, 0.10, 0.15, 0.20, 0.30, 0.40, 0.50 and 0.60. These results suggest that the designed nanoparticles are appropriate for solid-state lighting.” was added in the revised manuscript (Page 8 paragraph 2 line 12-17).

  1. “commercial NUV chip” Manufacturer, type?

Responding: Thank you very much for pointing out this. The commercial NUV chip is purposed from EPILEDS and its central wavelength is around 400 nm. Furthermore, the “a warm white-LED was designed via the utilization of a commercial NUV chip, which was bought from EPILEDS and its central wavelength and full width at half maximum are 400 and 14.4 nm, respectively,” was added in the revised manuscript (Page 9 paragraph 2 line 2-4).

Round 2

Reviewer 1 Report

The authors have responded in detail to the reviewer comments. Clarifications were provided and several modifications and additions were made to the manuscript. The quality of the manuscript improved. Nevertheless several issues remain:

1)      Regarding comment 1, it would have been informative to also have lattice constants derived for the different compositions (or at least for the end members), to assess the impact of x on the lattice constants.

2)      Regarding comment 5, for the cited references, the theta value is indeed ‘very close’ to 6. In the present manuscript it isn’t. So it would suggest to rephrase “Thereby, the θ value is estimated to be 6.87 and it is very close to 6” to “Thereby, the θ value is estimated to be 6.87, which is closer to 6 than to 8, ...”

3)      Regarding comment 6,  the newly added text contains a typo. “was employed to recorded the emission spectrum”… should be “was employed when recording the emission spectrum

4)      Regarding comment 7, the second part of the modified sentence in the manuscript “In this formula, I0 is the fluorescence intensity at time t, which is much longer than the decay time τ and I(t) are associated with the fluorescence intensities at time t = 0 and t, respectively, and A is ascribed to constant” is not correct. “In this formula, I(t) is the fluorescence intensity at time t, while I0 is the fluorescence intensity at times t much longer than the decay time τ.

5)      Regarding comment 8: the approach to calculate the refractive index from equation (6) is not valid, as this approach (following equations 4 and 5), implicitly assumes that there are no non-radiative transitions. However, from both the quantum efficiency measurements and the lifetime values (taken here as 4.41ms) it is clear that is not the case. For instance, a much higher lifetime is found for the sample with lowest x.  

6)      Regarding comment 9: as a consequence of the previous comment, the refractive index value of 1.92 seems to be too high for a fluoride compound (for instance, KF is 1.35, GdF3 is 1.59, NdGdF4 is about 1.50).

7)      Regarding comment 10: it is positive that the IQE and EQE have now been measured more accurately. However, it would be beneficial to add error margins and some comments. It is peculiar to see that going from x = 0.05 to about x = 0.3 the IQE increases, while at the same time the decay time decreases drastically. Furthermore, in the sentence “ “Excited by 393 nm, the IQE values of KGd2F7:2xEu3+ nanoparticles are determined to be 42.6%, 58.0%, 73.6%, 78.1%, 59.5%, 43.6% and 20.2%, respectively, when x = 0.05, 0.10, 0.15, 0.20, 0.30, 0.40, 0.50 and 0.60” there are 7 values for the IQE, whereas 8 compositions are mentioned.

8)      Regarding comment 14: it is positive that the sentence was rephrased. I would like to mention though that the figure R1 (only used in the rebuttal letter) is physically not correct. For Eu3+ there is no shift for the parabola of the 5D0 excited state as compared to the parabola for the 7F0 ground state (otherwise broadband emission would be found for Eu3+). For 4f-4f emittors, other quenching routes are found compared to the one mentioned in the rebuttal letter.

Author Response

Dear editor, dear reviewers:

We are pleased to get that our manuscript has received a positive review. Thank you and the reviewers for reviewing my manuscript and pointing out the significant objections. We think we have complied with the reviewers' comments in our revised manuscript. We hope our manuscript could be accepted this time.

Responding to the comments of Reviewer #1:

  1. Regarding comment 1, it would have been informative to also have lattice constants derived for the different compositions (or at least for the end members), to assess the impact of x on the lattice constants.

Responding: Thank you very much for your suggestion. In general, to estimate the lattice parameters of studied sample via their XRD patterns, the Rietveld XRD refinement is widely used. As for performing the Rietveld XRD refinement, the crystallographic information file (cif) of studied sample is required. Unfortunately, we do not have the cif of KGd2F7, which makes us difficult to do the Rietveld XRD refinement. Thus, it is impossible for us to supply the accurate lattice parameters of resultant nanoparticles at the moment. In present work, we tried to adopt the XRD technique to explore the phase compositions of designed nanoparticles. Based on the recorded results (see Figure 1), one knows that all of the prepared compounds have pure monoclinic phase which is independent on Eu3+ content.

  1. Regarding comment 5, for the cited references, the theta value is indeed ‘very close’ to 6. In the present manuscript it isn’t. So it would suggest to rephrase “Thereby, the θ value is estimated to be 6.87 and it is very close to 6” to “Thereby, the θ value is estimated to be 6.87, which is closer to 6 than to 8, ...”

Responding: Thank you very much for pointing out this. We have checked the manuscript carefully and this mistake has been corrected in the revised manuscript.

  1. Regarding comment 6,  the newly added text contains a typo. “was employed to recorded the emission spectrum”… should be “was employed when recordingthe emission spectrum”

Responding: Thank you very much for pointing out this. The aforementioned mistake has been corrected in the revised manuscript.

  1. Regarding comment 7, the second part of the modified sentence in the manuscript “In this formula, I0is the fluorescence intensity at time t, which is much longer than the decay time τ and I(t) are associated with the fluorescence intensities at time = 0 and t, respectively, and is ascribed to constant” is not correct. “In this formula, I(t) is the fluorescence intensity at time twhile I0 is the fluorescence intensity at times t much longer than the decay time τ.

Responding: Responding: Thank you very much for pointing out this. This above mistake has been modified in the revised manuscript.

  1. Regarding comment 8: the approach to calculate the refractive index from equation (6) is not valid, as this approach (following equations 4 and 5), implicitly assumes that there are no non-radiative transitions. However, from both the quantum efficiency measurements and the lifetime values (taken here as 4.41ms) it is clear that is not the case. For instance, a much higher lifetime is found for the sample with lowest x.  

Responding: Thank you very much for pointing out this. From previous literatures (J. Appl. Phys. 109 (2011) 053511; J. Mater. Chem. C 1 (2013) 2338), one knows that the total radiative transition probability can be expressed by using the radiative transition lifetime, as defined below:

                             (R1)

where τ0 refers to radiative decay time. In previous version, it is our mistake that we directly used the decay time to estimate the refractive index of studied samples. Actually, it should be the radiative transition lifetime, τ0. On the other hand, via the use of the Auzel’s model to analyze the relation between decay time and doping content, the τ0 value can be obtained, as described below (J. Lumin. 100 (2002) 125):

                (R2)

In this expression, τ(x) is the lifetime at dopant content of x, τ0 is assigned to the radiative transition lifetime, C0 is constant and N denotes the generated phonon number for quenching the studied level by means of the cascade multiphonon process. Thus, through fitting these experimental data with the aid of Eq. (4), the value of τ0 is found to be about 11.55 ms. As a consequence, by means of Eq. (5)-(8), one achieves that refractive index of studied samples is determined to be 1.39, whereas the values of Ω2 and Ω4 of Eu3+ in resultant nanoparticles are determined to be 1.06 × 10-20 and 2.18 × 10-20 cm-2, respectively. Ultimately, the “Furthermore, the relation between decay time and dopant content is able to be further analyzed by Aizel’s model [28]:

                        (4)

where τ(x) refers to the lifetime at dopant content of x, τ0 is assigned to the radiative transition lifetime, C0 is constant and N denotes the generated phonon number for quenching the studied level by means of the cascade multiphonon process. Apparently, through fitting these experimental data by Eq. (4), the value of τ0 is found to be about 11.55 ms.” was added in the revised manuscript (Page 7 paragraph 1 line 4-11).

  1. Regarding comment 9: as a consequence of the previous comment, the refractive index value of 1.92 seems to be too high for a fluoride compound (for instance, KF is 1.35, GdF3 is 1.59, NdGdF4 is about 1.50).

Responding: Thank you very much for pointing out this. As analyzed above, we calculated the refractive index of studied sample again and its value is found to be 1.39. Clearly, the calculated value matches well other developed fluoride compounds.

  1. Regarding comment 10: it is positive that the IQE and EQE have now been measured more accurately. However, it would be beneficial to add error margins and some comments. It is peculiar to see that going from x = 0.05 to about x = 0.3 the IQE increases, while at the same time the decay time decreases drastically. Furthermore, in the sentence ““Excited by 393 nm, the IQE values of KGd2F7:2xEu3+ nanoparticles are determined to be 42.6%, 58.0%, 73.6%, 78.1%, 59.5%, 43.6% and 20.2%, respectively, when x = 0.05, 0.10, 0.15, 0.20, 0.30, 0.40, 0.50 and 0.60” there are 7 values for the IQE, whereas 8 compositions are mentioned.

Responding: Thank you very much for your suggestion. In present work, the fluorescence spectrometer (Edinburgh FSL1000), which was attached with an integrating sphere, to measure the quantum efficiencies of studied samples. In terms of the Edinburgh FSL1000, it will directly present the internal quantum efficiency (IQE) of studied samples, as displayed in Figure R1. Moreover, to measure quantum efficiencies of resultant nanoparticles, we sent our samples to a testing organization and they only supplied one data for each sample. Furthermore, we do not have this device in our lab. Thereby, it is difficult for us to present the error margins at the moment. On the other hand, it has been widely reported that the decay time of Eu3+ is decreased with the increment of doping content (J. Am. Ceram. Soc. 102 (2019) 3823; RSC. Adv. 11 (2011) 12981; J. Alloys Compd. 852 (2021) 157074), which coincides well with our results. Besides, it has been also found that the decay time and quantum efficiency of rare-earth ions doped luminescent materials present different changing tendency with the increase of doping content, namely, the quantum efficiency increases as the doping content increase and reaches its maximum value at the optimal doping content, and then it starts to decrease with further elevating the dopant content (J. Alloys Compd. 43 (2017) 2720; RSC Adv. 9 (2019) 20742; J. Alloys Compd. 785 (2019) 53; J. Rare Earths. 40 (2022) 1682). Evidently, these previous reports match well with our results. In addition, it is our mistake that IQE value of KGd2F7:0.60Eu3+ nanoparticles is missing in previous version and it has been supplied in the revised manuscript.

Figure R1. The measurement of IQE of KGd2F7:0.60Eu3+ nanoparticles.

  1. Regarding comment 14: it is positive that the sentence was rephrased. I would like to mention though that the figure R1 (only used in the rebuttal letter) is physically not correct. For Eu3+ there is no shift for the parabola of the 5D0 excited state as compared to the parabola for the 7F0 ground state (otherwise broadband emission would be found for Eu3+). For 4f-4f emittors, other quenching routes are found compared to the one mentioned in the rebuttal letter.

Responding: Thank you very much for pointing out this. We will pay much attention to the thermal quenching process of Eu3+-doped luminescent materials in our future work.

Reviewer 2 Report

The paper may be published in the present form

Author Response

Dear editor, dear reviewers:

We are pleased to get that our manuscript has received a positive review. Thank you and the reviewers for reviewing my manuscript and pointing out the significant objections. We think we have complied with the reviewers' comments in our revised manuscript. We hope our manuscript could be accepted this time.

Responding to the comments of Reviewer #2:

  1. The paper may be published in the present form

Responding: Thank you very much for your positive recommendation.

Round 3

Reviewer 1 Report

The authors have made the required changes to the manuscript. Although it would have been useful to the reader if certain observations were given somewhat more context (e.g. regarding the refractive index, the quantum efficiencies...) within the manuscript with reference to literature on similar compounds, it is not strictly necessary, so, I assume the paper can now be accepted.

Author Response

Dear editor and reviewer:

We are pleased to get that our manuscript has received a positive review. Thank you for reviewing my manuscript and pointing out the significant objections. We think we have complied with the proposed comments in our revised manuscript. We hope our manuscript could be accepted this time.

Responding to the comments of Reviewer#1:

  1. The authors have made the required changes to the manuscript. Although it would have been useful to the reader if certain observations were given somewhat more context (e.g. regarding the refractive index, the quantum efficiencies...) within the manuscript with reference to literature on similar compounds, it is not strictly necessary, so, I assume the paper can now be accepted.

Responding: Thank you very much for your kindness recommendation.